# Diffusion Language Models Are Natively Length-Aware

## Abstract

Unlike autoregressive language models, which terminate variable-length generation upon predicting an End-of-Sequence (EoS) token, Diffusion Language Models (DLMs) operate over a fixed maximum-length context window for a predetermined number of denoising steps. However, this process is independent of the required response length, resulting in computational waste for the majority of short responses common in reasoning and chat tasks. To address this problem, we conjecture that the latent prompt representation contains sufficient information to estimate the required output length. We provide empirical evidence for this phenomenon and propose a zero-shot mechanism to dynamically crop the context window before generation begins, leading to fewer diffusion steps and substantial computational savings. We evaluate our approach on four benchmarks with diverse tasks—GSM8K (reasoning), HumanEval (code generation), IfEval (instruction following), and LongFormQA (question answering)—revealing massive efficiency gains at minimal performance impact. We report significant reductions in FLOPs across all tasks, with no statistically significant performance degradation, and significant performance improvements in 2 out of 4 tasks.

## 1. Introduction

Language generation with Large Language Models (LLMs) has been dominated by autoregressive models, which generate text sequentially, predicting one token at a time (Zou et al., 2023). While proven successful, this sequential mechanism inherently limits inference speed and brings various disadvantages, such as the inability to perform global refinements during the generation process and error propagation

[1]Anonymous Institution, Anonymous City, Anonymous Region, Anonymous Country. Correspondence to: Anonymous Author <anon.email@domain.com>.

Preliminary work. Under review by the International Conference on Machine Learning (ICML). Do not distribute.

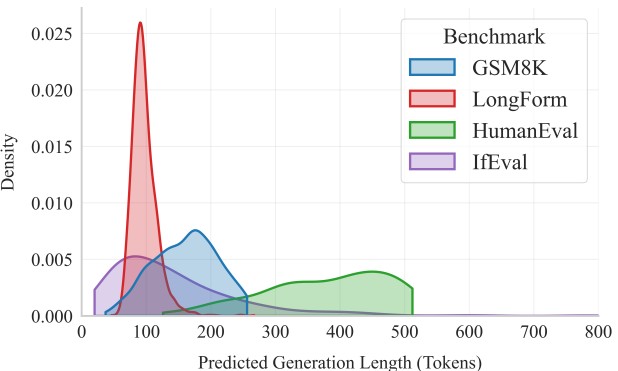

*Figure 1.* **Predicted Length Distributions.** Our SMARTCROP ($\tau = 0.9$) method successfully predicts task-specific output lengths across four benchmark datasets. The abrupt truncations observed in certain distributions correspond to context length constraints (refer to Section 4 for details).

due to "wrong" early tokens. Recently, Diffusion Language Models (DLMs, Austin et al., 2021; Lou et al., 2024) have emerged as a promising alternative, offering the potential for accelerated, non-autoregressive generation through iterative denoising DLMs operate by progressively unmasking tokens on a fixed-length canvas. This process is initialized with a prompt, while the remainder of the maximum context window is filled with placeholder mask tokens. In each denoising step, the model predicts logits for the entire masked sequence and unmasks a subset of tokens, typically those with the highest confidence. Unmasked tokens are kept and the process repeated. While this allows for flexible, parallel sequence generation, the requirement of a fixed canvas length remains a significant bottleneck, as the dimensions must be defined *a priori* based on heuristics or domain knowledge. To support variable-length outputs, current approaches often rely on padding with special End-of-Sequence (EoS) tokens to prevent unmasking beyond a certain point (Nie et al., 2025). However, this introduces substantial computational waste: the model must still process the entire context window during every forward pass, regardless of the actual output length.

In this work, we conjecture that DLMs implicitly encode the required output length within their latent representation of the prompt. In other words, DLMs encode an expectation about how many answer tokens are needed, depending on

the task or question they are prompted with. While these models are explicitly trained to predict EoS tokens at appropriate positions, we show that this length signal can be extracted and exploited *before* generation begins, rather than discovered iteratively during denoising.

Building on this, we introduce SMARTCROP, a model-native method to optimize DLM inference. We transform EoS logits into a cumulative "inverse survival" probability across positions, modeling the likelihood of the response terminating at any given point on the canvas. By identifying the first position where this probability exceeds a specified threshold (e.g., $\tau = 0.9$), we dynamically crop the canvas before generation begins. We then run the standard denoising schedule on the new, shorter canvas.

We evaluate our approach using LLaDA (Nie et al., 2025), a state-of-the-art 8 billion parameters DLM trained to handle variable-length outputs via EoS padding, and test it on four benchmarks with a diverse range of tasks: GSM8K (reasoning, Cobbe et al., 2021), HumanEval (code generation, Chen et al., 2021), IfEval (instruction following, Zhou et al., 2023), and LongFormQA (question answering Köksal et al., 2024).

Our results demonstrate that SMARTCROP drastically reduces computational costs, measured in FLOPs, without statistically significant performance degradation across any benchmark but significant performance improvements in 2 of the 4 evaluation suites. These findings suggest that DLMs trained with the EoS paradigm are inherently "length-aware", and that SMARTCROP effectively leverages this previously unobserved behavior to bridge the efficiency gap between fixed canvas diffusion and variable-length generation.

We make our code and results publicly available on GitHub.[1]

## 2. Related Work

DLMS typically decode by denoising a fixed-length canvas of length $L_c$ for $T$ steps, yielding an inference cost that scales roughly with $L_c \times T$ even when the desired output is short. We therefore situate SMARTCROP along two orthogonal axes explored by prior work: improving the *trajectory* (e.g., reducing steps $T$ or improving sampling quality) versus improving the *canvas allocation* (i.e., adapting $L_c$ to the prompt). We review (i) foundational diffusion methods for text, (ii) scaling diffusion to the LLM regime where the padding tax becomes practically significant, (iii) sampling-efficiency methods that reduce per-sample cost but keep $L_c$ fixed, (iv) diffusion-specific variable-length decoding that adapts length during or via retraining, and (v) length prediction ideas in other non-autoregressive LMs that motivate

---

[1] Code provided via zip file for review.

extracting termination signals from internal states.

**Diffusion Models for Text.** Diffusion models originate from non-equilibrium thermodynamics and were first developed for continuous data (Sohl-Dickstein et al., 2015; Ho et al., 2020). Several works adapt diffusion to discrete text. Structured Denoising Diffusion Models in Discrete State-Spaces (D3PM) define masked corruption processes for tokens and show that discrete diffusion can serve as a general generative framework (Austin et al., 2021). Diffusion-LM applies diffusion in the embedding space and improves controllable text generation (Li et al., 2022). DiffuSeq extends this idea to sequence-to-sequence tasks (Gong et al., 2023), and DiffusionBERT integrates diffusion training with pre-trained masked language models (He et al., 2023). Lou et al. (2024) model discrete diffusion by estimating ratios of the data distribution and obtain strong likelihood estimates. Zou et al. (2023) survey this growing line of work under the umbrella term diffusion language models. These works primarily address *modeling* and *generation quality*; they do not focus on inference-time waste from denoising large masked regions when the correct output is short, which is the bottleneck we target.

**Scaling.** Recent work scales DLMs to the LLM regime (DLLMs). Gong et al. (2025) adapt large autoregressive transformers into diffusion models and study how performance scales with model size. Liang et al. (2025) derive scaling laws for diffusion transformers and analyze compute–performance trade-offs. Nie et al. (2025) train a DLLM from scratch and shows that it can match autoregressive LLMs on instruction-following and reasoning benchmarks. These works demonstrate that DLLMs are competitive with autoregressive models, but they still rely on fixed-length diffusion canvases at inference time. Our method is designed to be orthogonal to scaling, without retraining or architectural changes.

**Efficiency and Sampling.** Several methods improve the sampling efficiency of diffusion models. Early work in vision reduces the number of denoising steps while maintaining sample quality (Ho et al., 2020). In text, DiffusionBERT reports that diffusion-style objectives can reuse pre-trained encoders and achieve strong generation quality with moderate sampling cost (He et al., 2023). Other work studies the trade-offs of non-autoregressive generation more broadly (Ren et al., 2020). These approaches make each denoising trajectory cheaper, but they keep the sequence length fixed and do not address the padding tax caused by short outputs. Block Diffusion instead interpolates between autoregressive and diffusion decoding (Arriola et al., 2025). It decodes overlapping blocks of tokens and can reduce latency for long sequences, but it does so at the cost of some of the full-sequence parallelism that makes DLMs attractive

and still relies on a predetermined maximum context length. In contrast, SMARTCROP reduces compute by shrinking the sequence length processed at every step (effective $L_c$), and is compatible with step-reduction methods since it targets a different axis of the $L_c \times T$ budget.

**Variable-Length Generation.** The most closely related line of work aims to remove the fixed canvas constraint. DAEDAL proposes a training-free method that dynamically adjusts the canvas length during sampling (Li et al., 2025). It starts from a short canvas, uses internal confidence scores to decide when to expand, and inserts additional masked tokens when parts of the sequence look under-developed. This strategy increases the fraction of useful tokens, but requires several rounds of expansion and custom heuristics.

Yang et al. introduce a diffusion LLM with native variable generation lengths (dLLM-Var) (Yang et al., 2025). They modify training so that the model predicts EoS more accurately and support blockwise diffusion guided by n EoS detection. At inference time, dLLM-Var can stop denoising when EoS is predicted with high confidence, without relying on a fixed context. This approach delivers large speedups but requires retraining with new objectives and specialized decoding.

Our work sits between these extremes. Like DAEDAL, we use a training-free method on top of existing scaled DLMs. Like dLLM-Var, we rely on EoS behavior. However, we do not expand or modify the canvas during denoising and we do not change training. Instead, we show that a single early denoising step already encodes a useful distribution over output length, and we exploit it to crop the canvas once before standard diffusion decoding.

**Length Control in Other LMs.** Non-autoregressive generation must manage output length despite parallel token prediction. Ren et al. (2020) analyze why non-autoregressive models lag behind autoregressive ones and show how knowledge distillation and source–target alignment can ease learning by reducing target-token dependency. Su et al. (2021) demonstrate that a pre-trained masked language model (BERT) can serve as a strong backbone for non-autoregressive text generation, and introduce mechanisms to mitigate both the inflexibility of prefixed output length and the conditional-independence assumption; they also propose a *ratio-first* decoding strategy for settings where output length can be approximately estimated. Kaneko & Okazaki (2023) reduce sequence length by predicting edit operations that remove redundant tokens.

Distillation and model compression reduce computational cost (Sanh et al., 2019), and early exiting has been explored in other settings. In contrast, our method targets diffusion-style decoding and uses an inverse survival function over EoS logits to decide how much of the canvas to keep before sampling.

## 3. Methodology

Let $V$ denote the vocabulary size, $L_p$ the length of the tokenized prompt, $L_{\text{new}}$ the maximum number of new tokens and $L_c = L_p + L_{\text{new}}$ the fixed context window size. At the initialization of the generation process, a DLM receives an input sequence $\mathbf{x}$, constructed by concatenating the raw tokenized prompt $\mathbf{x}_{\text{prompt}} = (x_1, \ldots, x_{L_p}) \in \{0, 1, \ldots, V-1\}^{L_p}$ with $L_{\text{new}}$ placeholder <mask> tokens:

$$\mathbf{x} = (\mathbf{x}_{\text{prompt}}, \underbrace{\texttt{<mask>}, \ldots, \texttt{<mask>}}_{L_{\text{new}} \text{ times}}) \qquad (1)$$

The model encodes this input into latent space and infers a probability distribution over the vocabulary for each masked position. Through iterative sampling, a subset of <mask> tokens is replaced (unmasked) at each step, and the process repeats until the sequence is fully generated. To support variable-length generation within this fixed-length paradigm, a special end-of-sentence token (EoS) is included in the vocabulary. The model learns to use EoS as a padding token for positions exceeding the meaningful output length. However, this architectural constraint forces the model to process the full context window of length $L_c$ during every forward pass, incurring significant and often unnecessary computational costs.

We hypothesize that during pre-training, the model learns to encode information regarding the required output length within the latent representation of the initial input. We propose to extract this signal to perform dynamic context truncation.

Let $L^*$ be a random variable representing the true output length (prompt plus generated tokens). We want to estimate the cumulative probability that the generation terminates at position $\ell$, i.e., $\Pr(L^* \leq \ell)$ for any $\ell \in \{L_p + 1, \ldots, L_c\}$. By applying a softmax function to the model's logits, we obtain the local probability $\phi_i = \Pr(\text{token}_i = \texttt{EoS})$ for each position $i$. Consequently, the probability that the sequence has *not* ended by position $\ell$ is the joint probability of not observing an EoS token at any position from $L_p + 1$ to $\ell$. Therefore, the cumulative probability of the sequence ending at or before $\ell$ is given by:

$$\Pr(L^* \leq \ell) = 1 - \prod_{j=L_p+1}^{\ell} (1 - \phi_j) \qquad (2)$$

Using this cumulative distribution, we determine the predicted length $\hat{L}$ as the minimal position where this probability exceeds a predefined confidence threshold $\tau \in [0, 1]$:

$$\hat{L} = \min\{\ell \in \{L_p + 1, \ldots, L_c\} \mid \Pr(L^* \leq \ell) \geq \tau\} \quad (3)$$

Upon determining $\hat{L}$, we truncate the initial context window by removing the final $L_c - \hat{L}$ `<mask>` tokens. This ensures that for all subsequent denoising steps, the model processes a reduced context window of size $\hat{L}$.

This method serves as a lightweight, plug-and-play optimization applied immediately after the initial forward pass, significantly reducing the computational burden for the remainder of the generation process.

# 4. Experiments

## 4.1. Models

Our experiments use `LLaDA` (Nie et al., 2025), a state-of-the-art 8 billion parameters DLM. It implements a masked denoising protocol within a fixed context window during baseline decoding and supports variable-length generation through the `EoS`-as-padding paradigm. Our evaluation focuses on this model as it is currently the only open-source, high-performance native DLM that incorporates this `EoS` capability.

While we also considered `ModernBERT` (Zhou et al., 2025) adapted for diffusion generation, the relatively small model scale and limited performance yielded inconclusive results. Consequently, we focus our analysis solely on `LLaDA`, as its architecture and training objective provide the most robust environment for evaluating "length-aware" behaviors in large-scale DLMs.

We emphasize that SMARTCROP is designed specifically for `EoS`-trained diffusion models and makes no claims about DLMs trained with alternative paradigms.

## 4.2. Benchmarks

We evaluate our proposed SMARTCROP method against baseline decoding across four capability benchmarks selected to span distinct output-length regimes. Although text generation tasks generally lack an explicit ground-truth length, this selection allows us to investigate length-prediction behavior across qualitatively different domains. As illustrated in Fig. 1, our method successfully recovers task-specific length distributions from the latent prompt representations.

To maintain consistency with established literature while exploring the limits of diffusion efficiency, we define a specific maximum number of new tokens ($L_{new}$) and number of denoising steps ($T$) for each task (Table 1).

**Mathematical Reasoning.** `GSM8K` targets structured, short-form reasoning (Cobbe et al., 2021). Following Nie et al. (2025), we set $L_{new} = 256$ tokens. Performance is measured via *Exact-Match Accuracy*, verifying if the final numerical output aligns with the ground truth.

| Benchmark | Max new tokens ($L_{new}$) | Steps ($T$) | Content |
|---|---|---|---|
| GSM8K | 256 | 256 | Math |
| HumanEval | 512 | 512 | Code |
| IfEval | 1280 | 1280 | Instruction |
| LongFormQA | 512 | 64 | QA |

*Table 1.* **Experimental Configurations.** Summary of hyperparameters across benchmark datasets. *Max new tokens ($L_{new}$)* denotes the number of masked tokens appended to the prompt in the diffusion canvas. *Steps ($T$)* indicates the total number of denoising iterations. For example, $L_{new} = 256$ with $T = 256$ implies a linear schedule where exactly 1 token is unmasked per step. *Content* describes the task domains represented in each dataset.

**Code Generation.** `HumanEval` evaluates functional correctness (Chen et al., 2021). We use $L_{new} = 512$ tokens, consistent with the original `LLaDA` evaluation. Performance is measured using *Pass@1*, which assesses the functional correctness of generated code via unit tests.

**Instruction Following.** `IfEval` assesses adherence to objective constraints (Zhou et al., 2023). We set the maximum number of new tokens $L_{new} = 1280$, following the experimental setup in Ye et al. (2025). This expanded window provides a testbed for our method's ability to achieve significant token savings. Performance is measured by *Strict Accuracy*, a prompt-level metric verifying if the response satisfies all specified constraints (e.g., formatting, word counts).

**Question Answering.** `LongFormQA` evaluates free-form answering (Köksal et al., 2024) and represents a realistic chat-based scenario. The maximum number of new tokens is set to $L_{new} = 512$ and, as detailed in Appendix B, the predicted length remains almost invariant regardless of $L_{new}$ in this setting. Performance is measured using *ROUGE-1*, quantifying the unigram overlap between the generation and the reference answer. We acknowledge that performance on this benchmark is sensitive to the metric's inherent dependence on total sequence length.

Given computational constraints, we adopted a focused parameter selection rather than an exhaustive grid search. For `GSM8K`, `HumanEval`, and `IfEval`, the number of denoising steps is set equal to the context length ($T = L_{new}$) to establish a high-fidelity baseline. For `LongFormQA`, we fix the number of denoising steps to $T = 64$ steps. This allows us to evaluate SMARTCROP in a high-density generation regime, where the model must predict multiple tokens per step.

## 4.3. Baselines

To validate our length estimation hypothesis, we compare two distinct decoding strategies.

*Table 2.* **Main Results.** Performance comparison between native diffusion decoding and our proposed dynamic cropping method across four benchmarks using the LLaDA architecture. *Benchmark* indicates the evaluation suite. *Method* distinguishes between the native Full Context (FC) baseline and our SMARTCROP (SC) approach at various length-prediction quantiles ($\tau$). $L_p$ represents the average prompt length in tokens (constant per benchmark). *Avg. Processed Length* denotes the mean number of total tokens processed by the model in each forward pass, comprising both the prompt and the generated sequence. For the FC baseline, this value is constant at $L_c = L_p + L_{\text{new}}$, where $L_p$ and $L_{\text{new}}$ represent the prompt length and the maximum sequence length, respectively. In contrast, for SC, the output length is defined by the predicted total length $\hat{L}$. *Metric* denotes the task-specific performance score. *FLOPs Saved* quantifies the reduction in total floating-point operations relative to the FC baseline (e.g., a 98% saving implies our method requires only 2% of the baseline computation). *Perf.* $\Delta$ reports the relative percentage change in *Metric* compared to the FC baseline. For all metrics marked with ↑, higher is better. We determine statistical significance via pairwise bootstrap hypothesis testing (5000 resamples) between FC and SC within each experimental condition. We compute paired differences on samples matched by document ID for both *Metric* and *FLOPs Saved*. We estimate two-sided $p$-values from the resulting bootstrap distributions. Significance levels are denoted as $^*p < 0.05,^{**} p < 0.01$, and $^{***}p < 0.001$.

| Benchmark | Method | $L_p$ | Avg. Processed Length | Metric ↑ | FLOPs Saved(%)↑ | Perf.$\Delta$(%)↑ |
|---|---|---|---|---|---|---|
| **IfEval** | FC | | 1367.2 | 0.4801 | - | - |
| | SC-0.5 | | 192.1 | 0.5342 | 98.47*** | +11.25* |
| | SC-0.75 | 87.2 | 208.0 | 0.5521 | 98.05*** | +14.99** |
| | SC-0.9 | | 222.0 | 0.5459 | 97.64*** | +13.70** |
| | SC-0.95 | | 230.5 | 0.5450 | 97.37*** | +13.50** |
| | SC-0.99 | | 243.8 | **0.5694** | 96.92*** | +18.58*** |
| **GSM8K** | FC | | 396.7 | **0.5616** | - | - |
| | SC-0.5 | | 239.2 | 0.5452 | 69.39*** | −2.92 |
| | SC-0.75 | 140.7 | 261.2 | 0.5516 | 59.09*** | −1.77 |
| | SC-0.9 | | 278.8 | 0.5457 | 50.15*** | −2.83 |
| | SC-0.95 | | 288.5 | 0.5490 | 44.93*** | −2.25 |
| | SC-0.99 | | 302.8 | 0.5520 | 37.01*** | −1.71 |
| **HumanEval** | FC | | 690.5 | 0.4592 | - | - |
| | SC-0.5 | | 488.2 | 0.4665 | 46.42*** | +1.59 |
| | SC-0.75 | 178.5 | 506.7 | 0.4688 | 41.06*** | +2.08 |
| | SC-0.9 | | 521.9 | **0.4851** | 36.53*** | +5.65 |
| | SC-0.95 | | 531.0 | 0.4598 | 33.98*** | +0.13 |
| | SC-0.99 | | 543.6 | 0.4106 | 30.16*** | −10.59 |
| **LongFormQA** | FC | | 589.6 | 0.1341 | - | - |
| | SC-0.5 | | 155.1 | 0.2115 | 85.40*** | +57.72*** |
| | SC-0.75 | 77.6 | 164.4 | 0.2152 | 82.56*** | +60.48*** |
| | SC-0.9 | | 172.7 | 0.2173 | 79.94*** | +62.01*** |
| | SC-0.95 | | 177.5 | 0.2196 | 78.35*** | +63.73*** |
| | SC-0.99 | | 185.2 | **0.2210** | 75.86*** | +64.83*** |

**Full Context.**   This serves as the standard diffusion baseline, where the model denoises a fixed-length diffusion canvas of size $L_{\text{new}}$ for all prompts.

**SMARTCROP.**   Our proposed mechanism dynamically adjusts the generation canvas. We perform a single forward pass at the initial denoising step and convert the EoS logits into a cumulative probability distribution over positions. The canvas is cropped at the smallest position where the cumulative probability exceeds a threshold $\tau \in [0, 1]$. Subsequent denoising steps are then executed exclusively on this reduced window.

We quantify computational efficiency using total floating-point operations (FLOPs) and report relative reductions compared to the *Full Context* baseline. Performance comparisons between the decoding strategies are detailed in Table 2.

## 4.4. Sensitivity Analysis

To evaluate the robustness of our length prediction $\hat{L}$, we perform a sensitivity analysis by perturbing the cropped window size. We modulate the effective context length by applying a deviation factor $\delta \in [-50\%, +50\%]$ to the predicted length $\hat{L}$, defined as:

$$\ell_\delta = \hat{L} \cdot (1 + \delta). \tag{4}$$

By sweeping $\delta$ in 10% increments, we observe the sensitivity of model performance to the allocated sequence length. This controlled perturbation allows us to discern whether the efficiency gains stem from the model's ability to identify a precise, prompt-specific bound, or if they are merely a byproduct of reducing a generic, oversized context.

Furthermore, to isolate the effect of task-specific length estimation, we introduce a stochastic control. We compare our method against a baseline where the context length

is sampled at random from the aggregate distribution of predicted lengths across the other benchmarks. This comparison serves to confirm that the observed performance is due to instance-specific predictions rather than a broad, task-agnostic reduction in sequence length.

We conduct this analysis on the `IfEval` benchmark, as its generous maximum diffusion canvas ($L_{new}$) provides a sufficiently wide range for meaningful observation. The results, illustrated in Fig. 2, demonstrate how model performance scales as the context window converges toward or diverges from our predicted optimum.

## 5. Results

### 5.1. SMARTCROP Substantially Reduces Compute Cost

SMARTCROP consistently reduces the fraction of the masked canvas the model must denoise at each iteration, which directly translates into fewer token-position evaluations per step, thereby lowering the total FLOPs count. Across our benchmarks, SMARTCROP reduces FLOPs by 46–98% relative to full-context diffusion decoding, achieving an average computational saving of 67%. The most substantial gains are observed in tasks requiring concise outputs, such as `IfEval`, `GSM8K`, and `LongFormQA`. In these settings, the predicted length distribution concentrates around approximately 200-250 tokens. Consequently, cropping eliminates vast unused regions of the canvas. This is particularly evident in `IfEval`, where we employ a longer $L_{new} = 1280$. Even on `HumanEval`, where the average output length is higher and almost saturates the available additional tokens, the method still yields compute savings, albeit through less aggressive cropping.

This pattern aligns with the desired behavior of an adaptive canvas: the mechanism allocates computational resources proportional to the complexity of the task, pruning the most when the output requirements are minimal.

Notably, on `LongFormQA` (where $T < L_{new}$), reducing the processed context length while maintaining a constant number of denoising steps results in fewer tokens unmasked per each step. We hypothesize that this partly drives the observed performance improvements in this task.

### 5.2. SMARTCROP Maintains and Often Enhances Output Quality

Counter-intuitively, the results presented in Table 2 demonstrate that dynamic context truncation typically stabilizes or enhances generation quality rather than degrading it. We initially hypothesized that aggressive cropping might introduce a strict Pareto trade-off between efficiency and accuracy, posing a risk of premature termination and leaving outputs incomplete. However, our findings refute this:

performance remains stable on `GSM8K` and `HumanEval`, while we observe significant improvements on `IfEval` and `LongFormQA`.

**Reasoning and Compact Contexts.** On `GSM8K`, we observe a substantial reduction of 52.11% FLOPs on average, with only a statistically insignificant performance degradation (2.3% on average). This behavior is consistent with the benchmark's characteristics: the maximum number of new tokens used in the baseline ($L_{new} = 256$) was already manually optimized to be highly compact, following the configurations from Nie et al. (2025). Consequently, dynamic cropping operates on a narrow margin, occasionally truncating reasoning chains essential for multi-step mathematical derivations. Nonetheless, the minor impact on accuracy is heavily offset by the halved computational cost, demonstrating the efficiency of our method even in highly optimized settings.

**Instruction Following and Hallucination Mitigation.** Conversely, on `IfEval`, where a generous context window was used to match the evaluation settings of Ye et al. (2025), SMARTCROP yields significant gains (+11% to +18%). We attribute this to the mitigation of degeneration issues common in diffusion models when using excessive padding. Large, sparse context windows can induce repetitive loops or "hallucinations" within the trailing empty space. By removing this surplus canvas, we hypothesize that the model's attention mechanism is sharpened, focusing strictly on relevant tokens rather than attending to noise or uninformative padding.

**Conciseness in Open-Ended Generation.** On `LongFormQA`, we record a sharp increase in *ROUGE-1* scores. While this metric naturally favors higher overlap-to-length ratios, this result highlights a beneficial property of our method: it enforces conciseness. By predicting a tighter output bound, the model avoids the verbose wandering often observed in fixed-length decoding, thereby improving information density.

**Functional Correctness in Code Generation.** Finally, results on `HumanEval` show minor, statistically insignificant fluctuations, suggesting that for code generation, our method provides a "free" efficiency boost without compromising functional correctness. We argue that in this setting, a shorter context window encourages the model to produce more concise yet effective code. Given the inherent variability in coding styles, SMARTCROP appears to bias the model toward simpler, more direct implementations without altering the logic required to pass test cases.

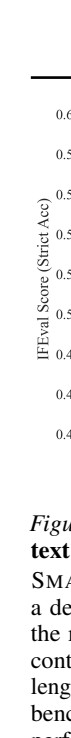
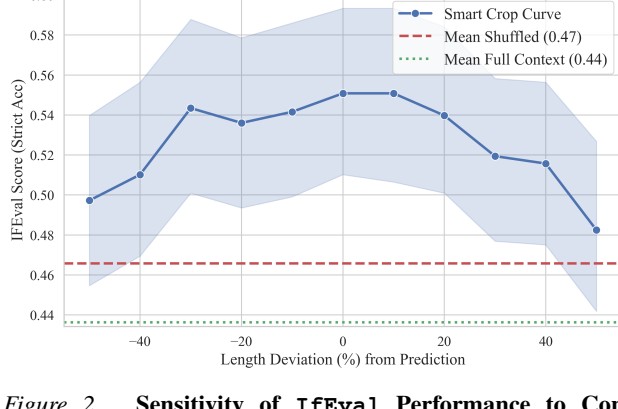

*Figure 2.* **Sensitivity of `IfEval` Performance to Context Length Perturbations.** We analyze the robustness of SMARTCROP ($\tau = 0.9$) by shifting the predicted length $\hat{L}$ by a deviation factor $\delta \in [-50\%, +50\%]$. The *blue curve* shows the model performance (mean $\pm$ 95% CI) across these varying context lengths. The *red line* denotes the control baseline, where lengths are sampled from the empirical length distribution of other benchmarks. The *green line* represents the *Full Context* baseline performance. While the model is relatively robust to moderate under-estimation (negative $\delta$), generation quality degrades as superfluous padding is reintroduced (positive $\delta$), eventually converging toward the baseline.

### 5.3. Sensitivity Analysis

Fig. 2 illustrates the *Strict Accuracy* scores across the perturbed context lengths. Our analysis reveals a distinct asymmetry in performance sensitivity relative to the predicted length $\hat{L}$.

**Robustness to Aggressive Cropping ($\delta < 0$).** Performance remains remarkably stable even when the context is cropped up to $20\%$ further than the initial prediction. However, once the additional cropping exceeds this threshold, accuracy drops sharply, nearly converging with the shuffled baseline. This suggests that the latent representations encode a conservative upper bound for the required computation, effectively providing a "safety margin" for the generation process. Tasks can often be successfully resolved within an even tighter window than $\hat{L}$ initially suggests.

**Degradation from Over-Padding ($\delta > 0$).** Conversely, extending the context window beyond the predicted length triggers an immediate decline in performance. As $\delta$ approaches $+50\%$, accuracy decreases from $0.48$ to $0.41$. This trend confirms that excess padding in DLMs is not merely computationally inefficient, but actively deleterious to generation quality.

**Baseline Comparison.** The SMARTCROP trajectory across the perturbed range consistently outperforms the *Full Context* baseline, which suffers from the noise inherent in the maximum fixed window, and the *Shuffled* control (in-

dicated by the red dashed line). This validates that our predicted lengths are truly instance-specific and provide superior guidance compared to a generic length prior.

These findings indicate that $\hat{L}$ serves as a robust "Goldilocks" threshold: it is sufficiently constrained to filter out the noise of an oversized context while remaining expansive enough to preserve the integrity of the output.

## 6. Conclusion & Discussion

DLMs typically operate on a fixed-length canvas, employing special `EoS` tokens as padding to accommodate variable-length sequences. While this design simplifies the training and sampling pipeline, it imposes a significant "padding tax" at inference time: the model must process the entire context window even when the majority of the sequence consists of redundant placeholder tokens. In this work, we demonstrate that this computational overhead can be avoided.

Specifically, we hypothesize that DLMs trained under the `EoS` paradigm, exemplified by the 8B-parameter `LLaDA` model, encode a usable length signal within the prompt's latent representation prior to the initiation of the denoising process.

To exploit this latent signal, we introduce SMARTCROP, a zero-shot, architecture-agnostic method. SMARTCROP extracts this information by transforming the initial `EoS` logits into an inverse survival probability function across sequence positions, thereby estimating the probability of sequence termination at each token index. The method identifies the optimal truncation point as the first position where this probability exceeds a predefined threshold, enabling standard diffusion decoding on a significantly shorter canvas. Critically, SMARTCROP requires no retraining, architectural modifications, or changes to the underlying decoder.

Our empirical evaluation confirms that SMARTCROP, at many different probability thresholds, effectively predicts required output lengths and yields substantial efficiency gains across diverse benchmarks. Remarkably, this reduction in computation does not compromise model performance. In fact, we observe statistically significant performance improvements in two out of four tasks, with no degradation in the remaining two. These results suggest that excessive padding is not merely computationally wasteful but potentially detrimental; it may destabilize the denoising process by encouraging degenerate behavior in empty regions of the canvas. By constraining the generation space, SMARTCROP shifts the model into a regime that favors more focused, higher-fidelity outputs.

Our sensitivity analysis further clarifies the nature of this length awareness. The distinct predicted length distributions observed across benchmarks indicate that the model

internalizes task-specific priors. Furthermore, the superior performance of SMARTCROP relative to shuffled controls suggests a sophisticated, prompt-conditioned understanding of sequence length.

These findings indicate that length awareness is a learned capability of DLMs trained with `EoS` padding, offering a promising trajectory toward more efficient and robust non-autoregressive generation.

## 7. Limitations

Despite the efficiency gains demonstrated by SMARTCROP, we identify three primary limitations of our work.

First, SMARTCROP introduces challenges for synchronized batch inference. Because the method dynamically adjusts the canvas size based on prompt-specific length predictions, different requests within a single batch may result in heterogeneous sequence lengths. This prevents straightforward hardware acceleration unless the inference engine implements specialized request grouping or padding strategies.

Second, the scope of our empirical evaluation is currently limited to a single diffusion architecture, `LLaDA` with `EoS` padding, and four English-language benchmarks. While our results are robust across these settings, the characteristics of the latent length signal may vary across different languages, specialized domains, or alternative decoding hyperparameters.

Third, the efficacy of SMARTCROP is based on the model's internal representation of `EoS` geometry. In models where the `EoS` token is poorly calibrated during pre-training, or in frameworks that omit an explicit termination token from the vocabulary entirely, the length signal may be less reliable. Extending our zero-shot mechanism to such architectures remains an open area for future research.

## 8. Future Work

The emergence of a latent length signal in DLMs opens several relevant directions for future research.

The most interesting to us would be investigating the temporal evolution of this signal during the denoising process. While we currently extract the length prediction at the first denoising step, we hypothesize that the accuracy of this estimation improves as the latent state converges toward a coherent sequence. However, using later denoising steps introduces a trade-off between predictive precision and the maximum achievable computational savings, a Pareto frontier that requires further exploration.

A second direction involves refining the extraction mechanism itself. While our current quantile-based heuristic is effective and zero-shot, replacing it with a lightweight,

learned projection layer could yield a more precise mapping from early latent states to the target length distribution. Such a predictor could potentially capture multi-modal length requirements that simple thresholding might overlook.

Finally, we anticipate that future DLM training procedures could explicitly optimize for length prediction accuracy. Beyond context cropping, this signal could be used to dynamically adapt the denoising schedule or trigger early exit conditions once the informative segments of the sequence have stabilized. Transitioning from fixed canvas decoding to such an adaptive, content-aware regime represents a significant step toward making diffusion-based generation as successful as its autoregressive counterparts.

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

## A. Experimental Setup

All experiments were conducted using the publicly available `GSAI-ML/LLaDA-8B-Instruct` checkpoint via the Hugging Face `transformers` library (Wolf et al., 2020). To optimize memory efficiency without compromising numerical stability, the model was loaded in `bfloat16` mixed precision. Benchmarking and inference evaluations were executed on high-performance computing (HPC) nodes, each equipped with four NVIDIA H100 NVL GPUs (94 GB HBM3).

## B. Sensitivity of Predicted Length Distributions

This section evaluates the sensitivity of the SMARTCROP length predictor to the *initial* canvas size $L_{\text{new}}$ used during the primary forward pass prior to cropping. For each benchmark, we maintain a fixed set of prompts and recompute the predicted response length $\hat{L}$ while varying $L_{\text{new}}$ across the range of values indicated in the legends. We report the *predicted new tokens*, defined as $\Delta\hat{L} = \hat{L} - L_p$, where $L_p$ denotes the prompt length.

As illustrated in the figures, the predicted length distributions exhibit significant overlap across varying initial context sizes, indicating that the length predictor is largely robust to the choice of $L_{\text{new}}$. Notably, for smaller values of $L_{\text{new}}$, the distribution exhibits a hard right-side truncation. This is a direct consequence of the constraint $\hat{L} \leq L_c$ (or equivalently, $\hat{L} - L_p \leq L_{\text{new}}$), which represents the physical upper bound imposed by the initial context window.

All results presented in this analysis are generated using a quantile threshold of $\tau = 0.9$.

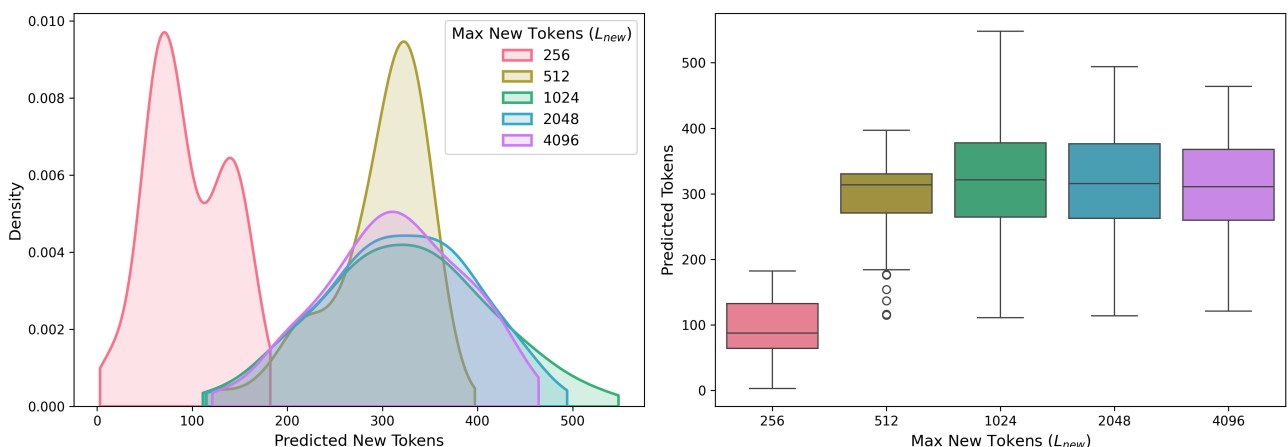

*Figure 3.* **Predicted Length Invariance (`HumanEval`).** Left: kernel density estimate of predicted new tokens for $L_{\text{new}} \in \{512, 1024, 2048, 4096\}$. Right: boxplots of the same values. The bulk of the distribution is comparatively stable across $L_{\text{new}}$, with the main visible difference being a stronger right truncation when $L_{\text{new}} = 512$, which is expected when the required completion length approaches the canvas limit. Note: $L_{\text{new}} = 256$ causes the predicted length distribution to be heavily truncated and uninformative compared to larger canvases.

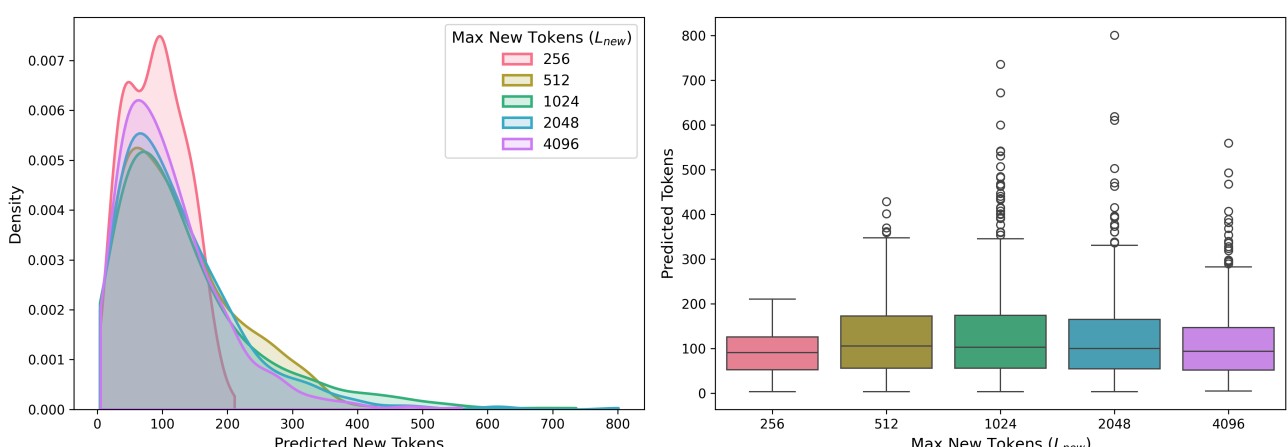

*Figure 4.* **Predicted Length Invariance (`IfEval`).** Left: kernel density estimate of predicted new tokens for $L_{\text{new}} \in \{256, 512, 1024, 2048, 4096\}$. Right: boxplots of the same values. The central mass of the predicted-length distribution (roughly 50–150 new tokens) is broadly consistent across $L_{\text{new}}$, while larger canvases primarily increase the range of rare long-length outliers.

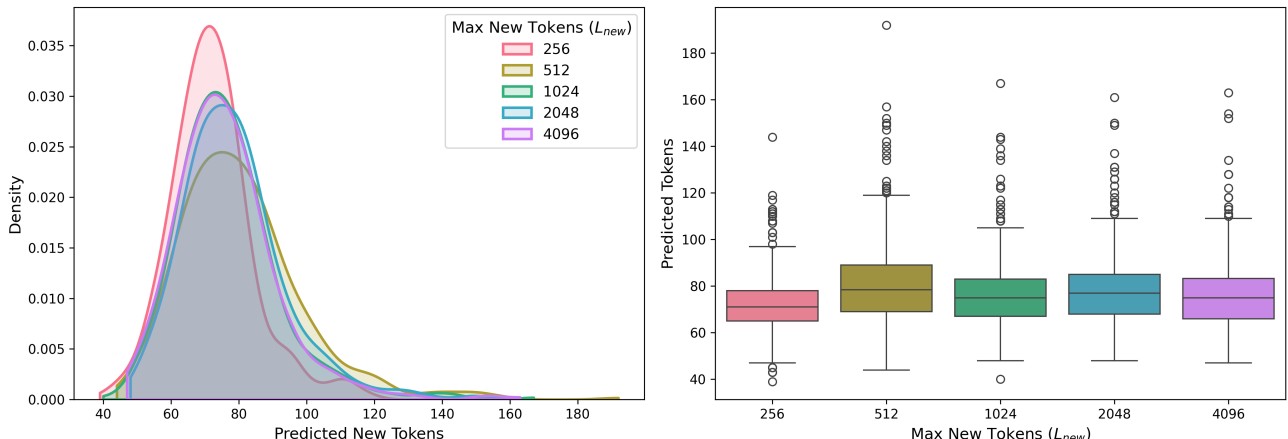

*Figure 5.* **Predicted Length Invariance (`LongFormQA`).** Left: kernel density estimate of predicted new tokens for $L_{\text{new}} \in \{256, 512, 1024, 2048, 4096\}$. Right: boxplots of the same values. The predicted length is close to invariant across $L_{\text{new}}$ for the typical range of outputs, with only small shifts in the median and dispersion. This supports the claim in the main text that, for `LongFormQA`, the model's inferred length prior is largely insensitive to the particular (potentially conservative) initial canvas size used for the first forward pass.

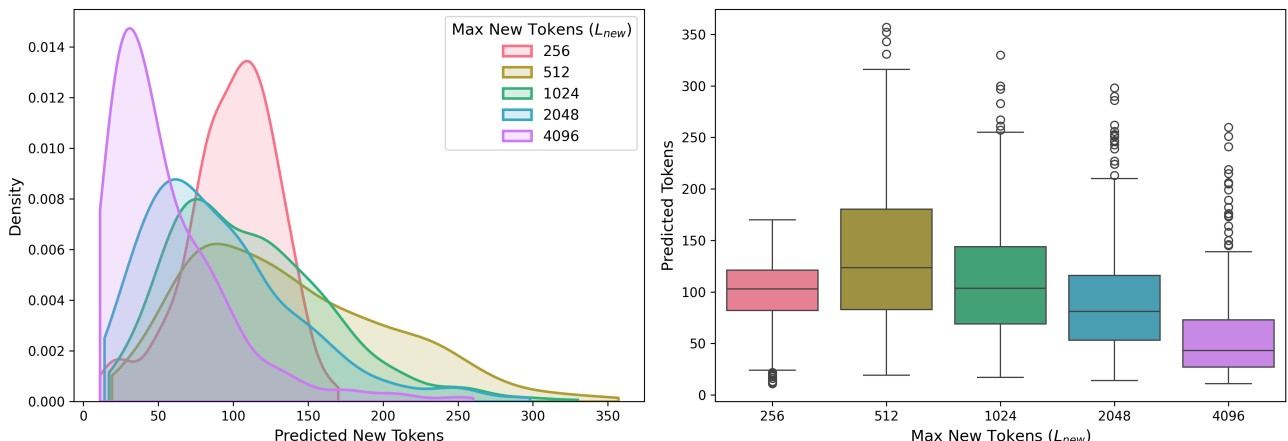

*Figure 6.* **Predicted Length Invariance (GSM8K).** Left: kernel density estimate of predicted new tokens for $L_{\text{new}} \in \{256, 512, 1024, 2048, 4096\}$. Right: boxplots of the same predicted token counts. We observe a pronounced dependence of the predicted length distribution on $L_{\text{new}}$ (most visible when setting $L_{\text{new}}$ at 256 or 4096), indicating that the per-position EoS probabilities used by SMARTCROP are not strictly invariant to the amount of masked padding presented to the model at the first denoising step. The $L_{\text{new}} = 256$ curve also shows a hard truncation consistent with the finite canvas constraint.

# C. Correlation between Predicted Length and Performance

This section investigates whether per-sample performance discrepancies between SMARTCROP and the *Full Context* baseline are systematically correlated with the predicted output length. Specifically, we examine whether performance gains are concentrated in instances with shorter predicted lengths. Such a correlation would suggest that improvements stem from a generic benefit of canvas reduction rather than our hypothesized mechanism of precise context window alignment.

Our analysis reveals no such systematic correlation, confirming that our method effectively identifies a precise, prompt-specific length. Figures 7, 8, 9, and 10 illustrate per-instance performance deltas (gray markers) relative to the generated token length. These are overlaid with a binned performance average (blue trend line) to highlight local trends.

For discrete metrics, *Exact Match* for GSM8K, *Pass@1* for HumanEval, and *Strict Accuracy* for IfEval, the per-example deltas $\Delta \in \{-1, 0, 1\}$. For LongFormQA, the delta is continuous. All plots correspond to a fixed SMARTCROP operating point with $\tau = 0.99$.

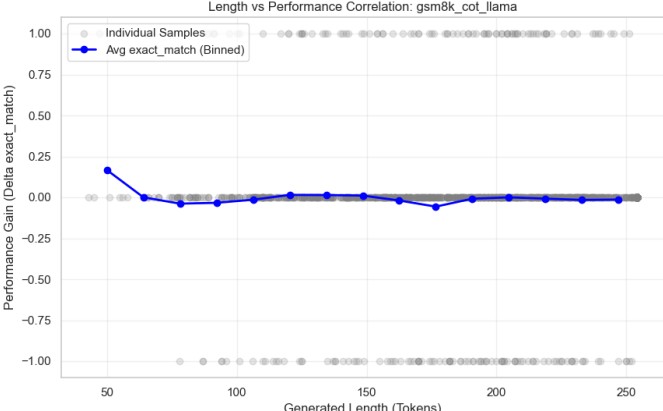

*Figure 7.* **Performance Gain vs. Length (GSM8K).** Per-instance change in *Exact Match* (grey; $\Delta \in \{-1, 0, +1\}$) plotted against generated length. The blue curve reports the mean delta within length bins.

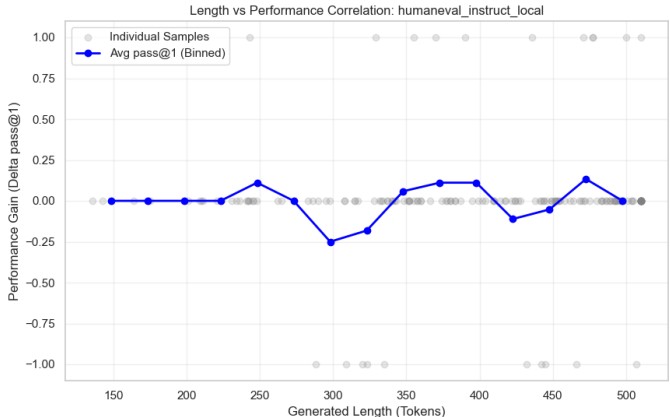

*Figure 8.* **Performance Gain vs. Length (HumanEval).** Per-instance change in *Pass@1* (grey; $\Delta \in \{-1, 0, +1\}$) plotted against generated length. The binned average (blue) is non-monotonic and fluctuates across bins, which is consistent with either weak dependence on length or limited sample counts per bin.

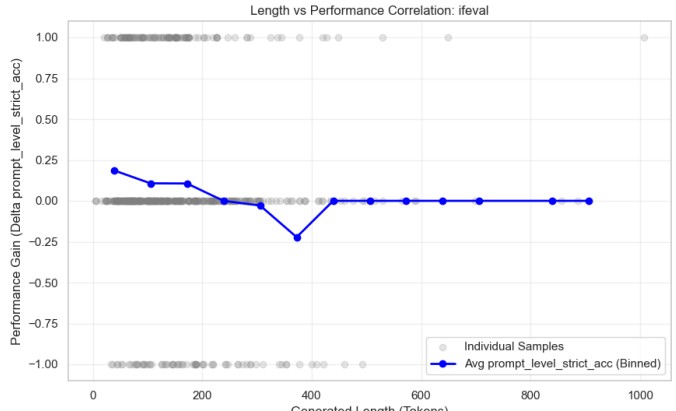

*Figure 9.* **Performance Gain vs. Length (`IfEval`).** Per-instance change in prompt-level *Strict Accuracy* (grey; $\Delta \in \{-1, 0, +1\}$) plotted against generated length. The binned average (blue) is positive for shorter generations and approaches zero for longer ones, indicating that the net gains from SMARTCROP at this operating point are concentrated on prompts that produce relatively short outputs. The negative dip around intermediate lengths should be interpreted with the corresponding bin sample size in mind.

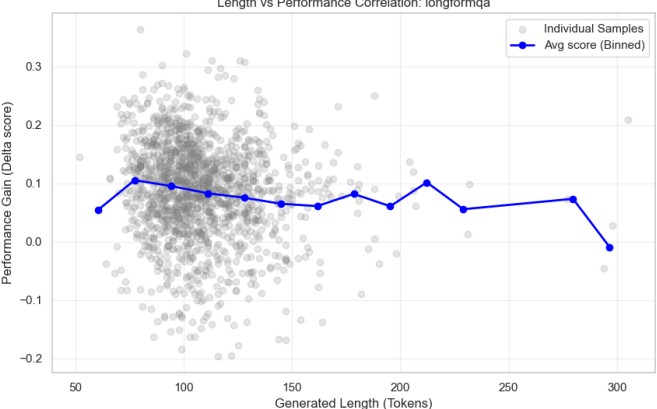

*Figure 10.* **Performance Gain vs. Length (`LongFormQA`).** Per-instance change in the evaluation score (grey; continuous delta) plotted against generated length. The binned mean delta (blue) is positive across the typical length range and does not exhibit a clear monotonic trend.

