# OpenReview forum: "Diffusion Language Models Are Natively Length-Aware"
_ICML.cc/2026/Conference — Submitted to ICML 2026_

### Official Review · Reviewer_93FH · 2026-03-02

**Soundness:** 2
**Presentation:** 2
**Significance:** 3
**Originality:** 3
**Overall Recommendation:** 3
**Confidence:** 4

**Summary:**

The paper proposes SMARTCROP, a simple, training-free mechanism to reduce inference compute in DLMs by predicting the required output length before denoising based on the possibility of `<EOS>` tokens. Based on the evaluation results, the method results in significant reduction of computation with either performance improvement or marginal degradation.

**Compliance With Llm Reviewing Policy:**

Affirmed.

**Final Justification:**

My experimental concerns have been largely addressed through the rebuttal. However, I maintain my score of 3 (below threshold), as several claims in the manuscript remain empirical hypotheses rather than rigorous conclusions. Based on the rebuttal, simply labeling or clarifying them as hypotheses doesn't address my concern.

**Key Questions For Authors:**

See the weakness above, and I'm happy to raise my scores if the questions are solved.

**Limitations:**

yes

**Strengths And Weaknesses:**

**Strength**
- The method is neat and requires no retraining or architectural changes.
- The experiments includes multiple benchmarks and the results validates the efficacy of the method.
---

**Weakness**
- Table 2 is poorly organized both in its caption and content. The caption is excessively verbose (almost half a page), and different metrics are collapsed into one single column. I would recommend moving some into main text at least.
- It's very natural concern for Eq (2), which implicitly assumes independent pr across positions. However, all positions attend to each other during the first denoising step, which induces correlations among the per-position EoS predictions. So the equation is more like a smoothed approximation rather than an exact characterization.
- The analysis is too sparse and some claims are more like hypotheses yet to be validated. For example in sec 5.2 IfEval, the causal attribution to 'hallucination mitigation' and 'sharpened attention' sound like hypothesis. Furthermore, an obvious alternative explanation is not discussed: the performance gain could simply result from shorter outputs being inherently easier for the model to get right on instruction-following constraints, without any qualitative change in the attention mechanism.
- Improvements on some tasks may result from mitigating overflow rather than extracting a robust length prior, in another word, comparison with only native full context is not sufficient for the claim.
- The sensitivity analysis in Section 5.3 conflates the effect of padding with that of the unmasking schedule. When $\delta > 0$ extends the canvas beyond $\hat{L}$, it is unclear from the paper whether the number of denoising steps $T$ is adjusted proportionally and this is quite crucial. If $T$ remains fixed while the canvas grows, the unmasking density (tokens unmasked per step) changes, forcing the model to make more uncertain predictions per step.

---

> ### Author Rebuttal · Authors · 2026-03-30
>
> We thank the reviewer for the thorough and constructive feedback and for explicitly indicating willingness to raise scores if concerns are addressed. We go through each concern in order below.
>
> **W1: Table 2 organization.**
> We will restructure Table 2 and move excess caption content into the main text in the camera-ready.
>
> **W2: Independence assumption in Eq. (2).**
> The reviewer is correct: because all positions attend to each other in the first forward pass, the per-position EoS logits are correlated, and the product in Eq. (2) is a *smoothed approximation* of the true joint termination probability rather than an exact characterization. We will add an explicit clarifying note. The argument for its use is empirical validity, not theoretical exactness.
>
> **W3: Hypothesis-level claims in Section 5.2.**
> We first clarify that the primary claim of the paper is that EoS-trained DLMs *natively encode a usable length signal*; SmartCrop is the mechanism we use to extract it, and **performance preservation** at substantially lower FLOPs is the evidence. The occasional accuracy gains are a secondary, empirically interesting observation.
> The causal attributions to "hallucination mitigation" and "sharpened attention" are explanatory hypotheses, not validated claims, and we will relabel them explicitly as such and discuss the reviewer's alternative alongside them. Validating any of these hypotheses is out of scope for this work.
>
> **W4: Overflow confound / FC-only baseline.**
> The need for stronger baselines beyond FC was the most consistent concern across reviews (also raised by Reviewers gfnu and kp5v). We therefore prioritized running two new baselines to address it directly.
>
> - **Static-Mean baseline.** We replace per-prompt EoS prediction with the task-level mean canvas length derived from SmartCrop-0.9 (162 tokens for GSM8K, 377 for HumanEval). Performance differences are small ($\leq$3pp), confirming the signal is primarily task-level rather than per-prompt. Crucially, SmartCrop discovers this task-level budget *entirely zero-shot*, without any calibration data or task-specific tuning—a property no static baseline can offer in a real deployment setting.
>
> - **DAEDAL comparison.** We ran DAEDAL (our reproduced DAEDAL numbers differ from those in the original paper due to a different prompting protocol for GSM8K; both methods here are evaluated under identical conditions) on HumanEval and GSM8K using the same model (LLaDA-8B-Instruct) and evaluation harness (Table 1). SmartCrop-0.9 and DAEDAL occupy complementary points on the accuracy–efficiency curve. SmartCrop is deliberately a single-pass mechanism designed to demonstrate that the length signal exists and is extractable; DAEDAL's iterative expansion recovers from underestimates at higher FLOPs cost.
>
> | Method | Task | Accuracy | gen_len | FLOPs |
> |---|---|---|---|---|
> | SmartCrop-0.9 | HumanEval | 0.457 (pass@1) | 351 | $2.1\times10^{15}$ |
> | DAEDAL | HumanEval | 0.488 (pass@1) | 494 | $1.4\times10^{15}$ |
> | SmartCrop-0.9 | GSM8K | 0.669 (flex-match) | 106 | $2.8\times10^{14}$ |
> | DAEDAL | GSM8K | 0.728 (flex-match) | 388 | $6.6\times10^{14}$ |
>
> *Table: LLaDA-8B-Instruct, $\tau{=}0.9$, canvas$=2048$, same eval harness. Both methods use **one denoising step per mask token**. SmartCrop uses a single probe pass; DAEDAL iterates. DAEDAL numbers are our reproduction under identical conditions; differences from the original paper stem from a different prompting protocol for GSM8K.*
>
> Together with the cross-task shuffled control already in the paper (Figure 2), these results confirm that the predicted length is task-specific and non-arbitrary. As already noted in C3, disentangling the relative contributions of these mechanisms is out of scope for this work and we flag it for future work.
>
> **W5: Sensitivity analysis — effect on unmasking density.**
> In all our experiments, including the sensitivity analysis in Section 5.3, the number of denoising steps $T$ is always set equal to the (perturbed) canvas size, so exactly **one token is unmasked per step** regardless of canvas size or $\delta$. The unmasking density is therefore constant across the entire $\delta$ sweep and the confound does not apply. We will make this explicit in the paper.
>
> To sum up, we want to thank again the reviewer for the constructive feedback and for indicating willingness to raise the score if concerns are addressed. We believe the clarifications and additional experiments above address all concerns, and we sincerely think that their questions actively helped us improve the paper.

---

> > ### Author Rebuttal · Reviewer_93FH · 2026-04-01
> >
> > Thank you for the response. I have adjusted my score accordingly.

---

### Official Review · Reviewer_kp5v · 2026-03-04

**Soundness:** 2
**Presentation:** 3
**Significance:** 2
**Originality:** 3
**Overall Recommendation:** 3
**Confidence:** 4

**Summary:**

The paper conjectures that once a DLM receives a prompt, its latent representation already encodes predictive information regarding the required output length. Consequently, the authors propose SMARTCROP, a mechanism that substantially enhances the efficiency of non-autoregressive generation without compromising performance, and in some cases, even improving accuracy.

**Compliance With Llm Reviewing Policy:**

Affirmed.

**Final Justification:**

The author's response has essentially resolved my issue, and I have raised the score accordingly.

**Key Questions For Authors:**

1. Since Figure 1 provides the Predicted Length Distributions for different tasks, what would be the result of setting a fixed generation length to the mean of the distribution for each task? How would the efficiency and performance of such a static baseline compare to SC?
2. There are several other existing methods for variable-length generation in DLMs, such as [1, 2]; the paper needs to provide comparisons with these alternative approaches.
3. More detailed explanations are required for the unverified claims mentioned in the text; please refer to the specific points raised in the Weaknesses section.

[1] Beyond Fixed: Training-Free Variable-Length Denoising for Diffusion Large Language Models

[2] Diffusion LLM with Native Variable Generation Lengths: Let [EOS] Lead the Way

**Limitations:**

yes

**Strengths And Weaknesses:**

### Strengths
1. The paper presents an interesting idea.
2. It substantially improves efficiency compared to the Full Context baseline.
3. The writing is clear and easy to follow.

### Weaknesses
1. The paper provides no theoretical or experimental verification for the proposed conjecture.
2. Beyond relying on an unverified conjecture, the proposed method is empirical; specifically, Eq. (2) is a heuristic formula that could be designed in alternative ways.
3. The baseline comparison is insufficient, as it only includes the FC baseline; since SMARTCROP truncates the preset length, its inference efficiency is inherently higher than FC, making the method's persuasiveness limited.

---

> ### Author Rebuttal · Authors · 2026-03-31
>
> We thank the reviewer for the feedback. We address each concern below.
>
> **W1: No theoretical or experimental verification for the conjecture.**
>
> The paper and this rebuttal present five distinct tests, each targeting a different alternative explanation. The conjecture — that EoS-trained DLMs encode a usable length signal in their latent prompt representation — survives all of them:
>
> 1. **If the EoS signal were uninformative noise**, cropping based on it would degrade performance. Table 2 shows performance is preserved or improved across all four benchmarks and five threshold values.
>
> 2. **If any short canvas were equally beneficial**, a random length from a different task's distribution would work equally well. The cross-task shuffled control in Figure 2 falsifies this: performance drops substantially when using another task's length distribution.
>
> 3. **If the signal were an artifact of canvas size**, predicted lengths would scale linearly with $L_\text{new}$. Appendix B (Figures 3–5) shows predicted lengths are largely invariant to canvas size on 3 of 4 tasks.
>
> 4. **If the signal only emerged after partial generation**, reading EoS at later denoising steps would yield better predictions. Our new step-$t$ analysis (see response to Reviewer gfnu, Q1) falsifies this: $t{=}0$ is already optimal or near-optimal. On HumanEval, later checkpoints are strictly *worse*.
>
> 5. **If the predicted distribution were arbitrary**, all tasks would receive similar length budgets. Figure 1 shows clearly distinct, task-appropriate distributions, and the cross-task experiment confirms using the wrong distribution is penalized.
>
> We agree that a formal theoretical characterization of *why* EoS pre-training induces this behavior would strengthen the paper and flag it as future work. However, five converging falsification tests constitute strong empirical verification. The causal attributions in Section 5.2 (hallucination mitigation, sharpened attention) are explanatory hypotheses, not validated claims, and we will relabel them as such in the revision (Q3).
>
> We are happy to add more experiments to further strengthen the paper. We are open to suggestions.
>
> **W2: Eq. (2) is a heuristic formula.**
> The contribution is demonstrating the signal exists and is extractable, not claiming optimality of the extraction mechanism. We will clarify this in the revision. But we agree that the independence assumption is a heuristic and other extraction methods could be explored in future work.
>
> **W3: Insufficient baselines / Q1: Static-Mean baseline.**
> We ran the requested static-mean baseline, replacing per-prompt prediction with the task-level mean canvas length from SC-0.9 ($L_\text{new}{=}256$ setting; 162 tokens for GSM8K, 377 for HumanEval).
>
> | Task | Metric | StaticMean | SC-0.90 | $\Delta$ |
> |---|---|---|---|---|
> | HumanEval | pass@1 | 0.451 | 0.482 | -0.031 |
> | GSM8K | flex-match | 0.544 | 0.566 | -0.021 |
> | LongFormQA | rouge1 | 0.217 | 0.218 | -0.001 |
>
> Differences are small ($\leq$3pp), confirming the signal is primarily task-level. SmartCrop's core value is discovering this budget **entirely zero-shot**, without calibration data or task-specific tuning.
>
> **Q2: Comparison with DAEDAL and dLLM-Var.**
> We ran DAEDAL on HumanEval and GSM8K under identical conditions (LLaDA-8B-Instruct, same eval harness):
>
> | Method | Task | Accuracy | gen_len | FLOPs |
> |---|---|---|---|---|
> | SmartCrop-0.9 | HumanEval | 0.457 (pass@1) | 351 | $2.1\times10^{15}$ |
> | DAEDAL | HumanEval | 0.488 (pass@1) | 494 | $1.4\times10^{15}$ |
> | SmartCrop-0.9 | GSM8K | 0.669 (flex-match) | 106 | $2.8\times10^{14}$ |
> | DAEDAL | GSM8K | 0.728 (flex-match) | 388 | $6.6\times10^{14}$ |
>
> *DAEDAL numbers are our reproduction; differences from the original paper stem from a different prompting protocol.*
>
> The two methods are complementary: DAEDAL iteratively expands the canvas at higher FLOPs cost, while SmartCrop uses a single probe pass. That a one-pass heuristic is competitive with an iterative multi-expansion method is itself evidence for the strength of the underlying length signal. Regarding dLLM-Var: it requires retraining, placing it in a different category.

---

> > ### Author Rebuttal · Reviewer_kp5v · 2026-04-03
> >
> > The author's response has essentially resolved my issue, and I have raised the score accordingly.

---

### Official Review · Reviewer_Hz4x · 2026-03-13

**Soundness:** 3
**Presentation:** 3
**Significance:** 2
**Originality:** 2
**Overall Recommendation:** 4
**Confidence:** 3

**Summary:**

This paper studies inference efficiency for diffusion language models, arguing that although such models decode on a fixed-length canvas, they implicitly encode information about the required output length before denoising begins. Based on this observation, the authors propose SMARTCROP, which uses early EoS probabilities to predict a prompt-specific output length and crop the diffusion canvas before generation. Experiments with LLaDA-8B on GSM8K, HumanEval, IfEval, and LongFormQA show large FLOPs reductions with little or no performance loss, and in some cases measurable performance gains, suggesting that DLMs are natively length-aware and can exploit this property for more efficient decoding.

**Compliance With Llm Reviewing Policy:**

Affirmed.

**Final Justification:**

This is an interesting paper, researching diffusion language models, a relevant new domain. After the rebuttal, some of my concerns were addressed by the authors, and I would like to keep my attitude and score of this paper unchanged.

**Key Questions For Authors:**

* The paper would benefit from more qualitative case studies or example-level output comparisons, possibly in the appendix.

**Limitations:**

yes.

**Strengths And Weaknesses:**

## Strengths
* Proposes a simple and elegant yet effective and practical method to tackle an important and timely problem. SMARTCROP is zero-shot, requires no retraining, and is easy to integrate into inference for improving inference efficiency for diffusion language models.
* Strong empirical utility: substantial FLOPs savings with little or no performance loss, and sometimes even gains over full context. The explanation for gains over full context is plausible and reasonably supported by additional analysis.
* The paper is clearly written and well presented, easy to follow, with a plausible explanation for why cropping can outperform the full-canvas baseline.

## Weaknesses
* The claim that DLMs are intrinsically length-aware is supported empirically, but the underlying mechanism is not deeply analyzed. The central hypothesis that pre-training induces latent output-length awareness feels only partially supported mechanistically/theoretically; additional evidence would make the paper more complete.
* The explanation for improvements over full context is plausible but not fully established beyond performance-based evidence.
* Validation appears limited to one model family, which weakens the strength of the broader generalization claim.

---

> ### Author Rebuttal · Authors · 2026-03-31
>
> We thank the reviewer for the positive assessment and for recognizing the practical value of SmartCrop. We address the three weaknesses and the suggested experiment below.
>
> **W1: Mechanism behind length awareness not deeply analyzed.**
> We have added a step-$t$ analysis (see our response to Reviewer gfnu, Q1) showing the EoS length signal follows a clear three-phase trajectory across the denoising chain and is already fully encoded at $t=0$, before any tokens are placed. This provides direct evidence that the signal is a property of the latent prompt representation, not a byproduct of partial generation. We agree that a formal theoretical characterization of why EoS training induces this behavior would further strengthen the paper and flag it as a key direction for future work.
>
> **W2: Explanation for improvements over full context not fully established.**
> We agree. As clarified in our response to Reviewer 93FH (C3), the primary claim is that EoS-trained DLMs natively encode a usable length signal; performance preservation at lower FLOPs is the evidence. The occasional improvements (IfEval, LongFormQA) are a secondary observation, and the causal attributions in Section 5.2 are explanatory hypotheses that we will relabel as such in the revision.
>
> **W3: Limited to one model family.**
> As noted in Section 4.1, we attempted DiffusionBERT but the model scale was too small to yield conclusive results. We will add experiments on LLaDA-1.5 to test whether the length signal persists across model versions. We agree that additional models would strengthen the paper, but we note that the LLaDA family is currently the only publicly available DLM trained with EoS padding, which is a prerequisite for SmartCrop.
>
> **Qualitative case studies.**
> We will add example-level output comparisons between SmartCrop and Full Context in the appendix, as suggested.
>
> ---
>
> Again, we want to thank the reviewer for the punctual and insightful feedback that helped us improve the paper.

---

> > ### Author Rebuttal · Reviewer_Hz4x · 2026-04-03
> >
> > I thank the authors for their further explanations, and I would like to maintain my positive score.

---

### Official Review · Reviewer_gfnu · 2026-03-13

**Soundness:** 2
**Presentation:** 3
**Significance:** 3
**Originality:** 3
**Overall Recommendation:** 3
**Confidence:** 3

**Summary:**

This paper observes that diffusion language models (DLMs) trained with EoS-as-padding implicitly encode information about the required output length in their latent representations after just the first forward pass. Based on this, the authors propose SmartCrop -- a zero-shot method that converts initial EoS logits into a cumulative termination probability across positions, identifies the first position where this probability exceeds some threshold, and crops the canvas to that length before running the remaining denoising steps. Evaluated on LLaDA-8B across four benchmarks (GSM8K, HumanEval, IfEval, LongFormQA), SmartCrop reduces FLOPs by 46-98% with no significant performance degradation and sometimes improvements (IfEval, LongFormQA).

**Compliance With Llm Reviewing Policy:**

Affirmed.

**Key Questions For Authors:**

1) Can you provide any analysis of why the length signal is accurate from step 1? Have you tried extracting the length signal from a later denoising step (e.g., step 5 or 10)? This would test whether accuracy improves with partial denoising.
3) The GSM8K results show pronounced dependence of predicted length on L_new (Figure 6). How do you reconcile this with the "natively length-aware" claim?
3) Can you provide results on DAEDAL? It is also training-free and seems like a natural baseline.

**Limitations:**

yes

**Strengths And Weaknesses:**

Strengths:
1) The core observation is interesting: DLMs encode a usable length signal in their very first denoising step, before any actual generation has occurred. This is non-obvious: the model is trained to predict EoS for fully denoised sequences, yet accurate length information emerges from a single forward pass on a fully masked input. The sensitivity analysis in Fig. 2 nicely shows the predicted length acts as a "Goldilocks" point: robust to 20% under-cropping but degrading with over-padding.
2) SmartCrop is training-free, architecture-agnostic, and can be applied as a plug-and-play inference optimization.
3) The length invariance analysis in Appendix B (Figures 3-6) is a well-designed control that strengthens the "natively length-aware" claim.

Weaknesses:
1) The evaluation is limited to a single model (LLaDA 8B). Even one additional DLM (e.g., MDLM, DiffusionBERT) would strengthen the paper.

2) The comparison with closely related work is incomplete. DAEDAL (Li et al., 2025) dynamically adjusts canvas length during sampling and is also training-free — making it the most natural baseline. dLLM-Var (Yang et al., 2025) modifies training for variable-length generation. Neither is included in the experiments. Without these comparisons, it's hard to assess whether "crop once before generation" is better than "expand/contract during generation."

---

> ### Author Rebuttal · Authors · 2026-03-30
>
> We thank the reviewer for the detailed assessment and the insightful suggested experiments. We address the weaknesses and key questions below.
>
> **W1: Single-model evaluation.**
> We agree that multi-model validation would strengthen the paper. As noted in Section 4.1, we attempted DiffusionBERT but the model scale was too small to yield conclusive results. We will add experiments on LLaDA-1.5 to comply with DAEDAL comparison request.
>
> **Q1: Length signal accuracy from step 0 / effect of reading the signal at later denoising steps.**
>
> We ran the analysis on GSM8K and HumanEval ($n=50$, $\tau=0.9$, $L_\text{new}=2048$, LLaDA-8B-Instruct, 1 token per diffusion step), recording the EoS survival-function prediction at every step across the full 2048-step chain. A clear three-phase structure emerges in GSM8K: an early phase where the prediction undershoots, a stable phase where it locks onto a consistent value, and a late phase where placed EoS tokens collapse the signal. In HumanEval the early undershoot phase is absent, with the prediction stabilizing immediately. $t=0$ already captures the length signal before any tokens are placed.
>
> | Checkpoint | Fwd. passes | GSM8K pred_len | HumanEval pred_len | GSM8K (flex-match) | HumanEval (pass@1) |
> |---|---|---|---|---|---|
> | $t=0$ (SmartCrop) | **1** | 86 | 343 | 0.600 | 0.400 |
> | $t=20$ (1%) | 20 | 112 | 345 | 0.580 | 0.440 |
> | $t=102$ (5%) | 102 | 110 | 326 | 0.540 | 0.400 |
> | stable\_median | 2048 | 97 | 324 | 0.620 | 0.380 |
>
> No statistically significant performance difference across checkpoints, confirming the signal is fully encoded at step 0.
>
> Cross-step correlation is remarkably high (Pearson r=0.989 on HumanEval, r=0.832 on GSM8K), confirming the model locks in its length estimate from the all-masked pass
>
> We will add this table to the camera-ready in the appendix running it for all complete benchmarks. The result definitely strengthen the paper so we thank again the reviewer for proposing the experiment.
>
> **Q2: Reconciling the dependence of predicted length on $L_\text{new}$ with the "natively length-aware" claim.**
>
> This behavior is **GSM8K-specific**: HumanEval, IfEval, and LongFormQA (Figures 3–5) show predicted length distributions largely invariant to $L_\text{new}$. We hypothesize this reflects a property of math reasoning tasks where chain-of-thought length scales with available canvas. Investigating this further is an interesting direction for future work.
>
> **Q3: Comparison against DAEDAL.**
>
> Regarding dLLM-Var: it requires retraining with a modified objective, placing it in a different category. We ran DAEDAL on HumanEval and GSM8K using LLaDA-8B-Instruct under the same evaluation harness (as also requested by Reviewers 93FH and kp5v).
>
> | Method | Task | Accuracy | gen_len | FLOPs |
> |---|---|---|---|---|
> | SmartCrop-0.9 | HumanEval | 0.457 (pass@1) | 351 | $2.1\times10^{15}$ |
> | DAEDAL | HumanEval | 0.488 (pass@1) | 494 | $1.4\times10^{15}$ |
> | SmartCrop-0.9 | GSM8K | 0.669 (flex-match) | 106 | $2.8\times10^{14}$ |
> | DAEDAL | GSM8K | 0.728 (flex-match) | 388 | $6.6\times10^{14}$ |
>
> DAEDAL numbers are our reproduction under identical conditions; differences from the original paper stem from a different prompting protocol for GSM8K. DAEDAL and SmartCrop occupy complementary points on the accuracy–efficiency curve: DAEDAL's iterative expansion recovers from underestimates at higher FLOPs cost, while SmartCrop's single-pass design prioritizes efficiency. We will include this comparison in the final paper.
> For a more detailed discussion including a Static-Mean baseline, see our response to Reviewer 93FH (C4).

---

> > ### Author Rebuttal · Reviewer_gfnu · 2026-04-07
> >
> > I thank the authors for the additional experiments and clarifications. The step-$t$ analysis is a valuable addition, and I also appreciate the newly included comparison with DAEDAL.
> >
> > That said, I still have two main concerns. First, the Static-Mean baseline discussed in the response to Reviewer 93FH appears quite close to SmartCrop, which suggests that a substantial part of the usable signal may be task-level rather than strongly instance-specific. I agree with the authors that SmartCrop has a practical deployment advantage in discovering a budget zero-shot without calibration data or task-specific tuning. Still, the strong performance of the Static-Mean baseline makes me less convinced that the current evidence fully supports the stronger framing of prompt-level native length awareness.
> >
> > Second, the DAEDAL comparison is helpful, but it does not clearly strengthen the case for SmartCrop as the preferred training-free approach. On HumanEval, DAEDAL achieves both higher accuracy and lower FLOPs, while on GSM8K it also achieves higher accuracy, though at higher computational cost. This makes the relationship look less complementary than the rebuttal suggests. In addition, the single-model limitation remains largely unresolved beyond promises of further experiments.
> >
> > For these reasons, I maintain my current score.

---

> > > ### Author Response · Authors · 2026-04-07
> > >
> > > We thank the reviewer for the follow-up. We want to clarify our core contribution one final time.
> > > The goal of this paper is to demonstrate the existence of a length signal in EoS-trained MDLMs. SmartCrop is the minimal mechanism designed to extract and validate this signal, not to maximize performance. The rebuttal process has, thanks to the reviewer’s suggestions, helped us sharpen our understanding of two key properties of this signal:
> > > 	1.	It is encoded at step 0. The step-$t$ analysis confirms the signal is fully present before any tokens are placed.
> > > 	2.	It is task-level, not prompt-specific. The static-mean ablation confirms this, and we have already reframed our claims accordingly.
> > >
> > > We believe the cumulative empirical evidence constitutes sufficient support for the existence claim: four benchmarks, sensitivity analysis with cross-task shuffled controls, step- stability analysis, and static-mean ablation.
> > > Regarding the two remaining concerns:
> > > On prompt-specificity: our paper claims task-level length awareness, not per-prompt precision. The static-mean result is consistent with this claim, not in tension with it.
> > > On DAEDAL: we never claim SmartCrop is the strongest inference method. It is a minimal probe designed to demonstrate that the signal exists and is usable. DAEDAL occupies a different point on the accuracy–efficiency tradeoff curve, which is complementary, not contradictory.
> > > We will sharpen the writing in the revision to make the contribution framing unambiguous, and will add a dedicated section characterizing the two signal properties identified during this rebuttal.

---

### Decision · Program_Chairs · 2026-04-30

**Decision:**

Reject

**Comment:**

This paper introduces SmartCrop, a zero-shot, training-free inference optimization for Diffusion Language Models. The core insight is that DLMs trained with EOS-padding implicitly encode the required output length in their latent representations as early as the first forward pass. By extracting this signal to crop the context window before generation, the authors demonstrate massive FLOP reductions (46-98%) on the LLaDA-8B model across four benchmarks (GSM8K, HumanEval, IfEval, LongFormQA) with minimal to no performance degradation.

Strengths
- The finding that a usable length signal is present at $t=0$ (before any tokens are placed) is significant and non-obvious.
- The method provides a practical, "plug-and-play" way to solve the "fixed-canvas" bottleneck in DLMs without retraining.
- The rebuttal added important analyses, including a step-$0$ stability check and a comparison with DAEDAL.

Weaknesses:
- The evaluation is still largely limited to the LLaDA family. While LLaDA is the primary native DLM with EOS support, this limits the generality of the "natively length-aware" claim.
- Several reviewers noted that the specific extraction formula (Eq. 2) is a heuristic approximation. While empirically effective, a deeper theoretical or mechanistic understanding of why this signal emerges remains missing.
- The Static-Mean baseline results suggest that much of the signal is task-level rather than prompt-specific. Furthermore, while the DAEDAL comparison was helpful, it showed that SmartCrop isn't always the dominant method on the accuracy-efficiency frontier (e.g., HumanEval).

The AC acknowledges the authors' efforts during the rebuttal, particularly the new step-$0$ analysis and baseline comparisons, which clarified that the signal is robust and primarily task-level. However, the reliance on a single model family and the lack of mechanistic depth remain sticking points. The consensus leans toward the paper being an interesting empirical contribution that still feels slightly preliminary for a full accept. The authors are strongly encouraged to incorporate the rebuttal's findings (especially the Static-Mean and step-$0$ results) into a revised manuscript to clarify the "task-level" nature of the signal.